# Transmissive Polarizer Metasurfaces: From Microwave to Optical Regimes

**DOI:** 10.3390/nano12101705

**Published:** 2022-05-17

**Authors:** Ayesha Kosar Fahad, Cunjun Ruan, Rabia Nazir, Bilal Hassan

**Affiliations:** 1School of Electronics and Information Engineering, Beihang University, Beijing 100191, China; ayeshakosar@buaa.edu.cn; 2Beijing Key Laboratory for Microwave Sensing and Security Applications, Beihang University, Beijing 100191, China; 3Faculty of Electrical Engineering, University of Engineering and Technology, Lahore 100191, Pakistan; rabiamalkana@hotmail.com; 4Department of Electrical Engineering and Computer Science, Khalifa University of Science and Technology, Abu Dhabi 127788, United Arab Emirates; bilz@live.com

**Keywords:** MSs, polarization converters, polarizers, multiband transmissions, configurable converters

## Abstract

Metasurfaces, a special class of metamaterials, have recently become a rapidly growing field, particularly for thin polarization converters. They can be fabricated using a simple fabrication process due to their smaller planar profile, both in the microwave and optical regimes. In this paper, the recent progress in MSs for linear polarization (LP) to circular polarization (CP) conversion in transmission mode is reviewed. Starting from history, modeling and the theory of MSs, uncontrollable single and multiple bands and LP-to-CP conversions, are discussed and analyzed. Moreover, detailed reconfigurable MS-based LP-to-CP converters are presented. Further, key findings on the state-of-the-arts are discussed and tabulated to give readers a quick overview. Finally, a conclusion is drawn by providing opinions on future developments in this growing research field.

## 1. Introduction

According to Huygen’s principle [1], each infinitesimally small area of a confined surface has an associated secondary-field source. This means that fields around the surrounding secondary-field sources can be controlled by either changing the source or by changing the electromagnetic properties of the confined surface. It depends upon whether one is interested in the transmitted properties of the confined surface or reflected properties or both. Thus, any desirable properties can be achieved by the proper design of such surfaces. This process is often termed as shaping electromagnetic waves. By shaping, we mean either amplitude shaping or polarization shaping. Dielectric lenses, metallic mirrors, and reflectors are widely known for shaping electromagnetic waves in optics and microwave antenna engineering [2,3]. However, both solutions for shaping are bulky and often have considerable size, weight, and volume even in a millimeter-wave band. Another way to shape electromagnetic waves into or out of the confined surface is to split the surface into multiple elements, each having a polarizable inclusion with specifically selected parameters. This concept, which had been used in reflect and transmit arrays [4,5] was applied to Fresnel and chirped lenses, and frequency-selective surfaces [6].

Recent progress in metamaterials (MMs) revealed that there exists a profound and sophisticated class of materials that can shape electromagnetic waves by a design of subwavelength inclusions arranged in densely packed 3D structures. Such inclusions owing to specific electric and magnetic polarizations can control transmitted and reflected fields which in turn offer intriguing opportunities for microwave, millimeter, and optical regions [7,8,9]. However, these materials suffer from several disadvantages. Firstly, fabrication of bulk MM is very complex, particularly when they lie in the optical region, as it requires bulk MM to be 3D nanostructures. Moreover, wave propagation in metamaterials covers a substantial distance, therefore, they encounter high ohmic losses.

Recently, it was realized that 2D metamaterials can be a more profound type of metamaterial, having the ability to shape/control electromagnetic waves. They have been termed MSs (MS). They can have subwavelength inclusions to tailor electric and magnetic fields, to control transmitted and reflected fields. They are ideal candidates for many novel microwave and optical devices.

The basic properties of MSs are defined by their elementary subwavelength patterns, which result in unusual resonant behavior. Importantly, MSs are patterned with subwavelength structures, so homogeneous or nearly homogenous MSs transmit and reflect plane waves. Since MSs are thin structured and 2D in nature, they are less lossy and easier to fabricate. Another major advantage of using MSs is their easier integration with existing microwave and nanophotonics systems. Several factors influencing the applications and properties of MSs are the type of structure/pattern, their mutual/cross-couplings between adjacent cells, and the substrate used. The most attractive feature of the MS is its ability to control reflected and transmitted fields, which makes it highly effective for certain applications and replaces conventional bulky equipment. There have been excellent review articles on MSs covering theory, fundamentals and applications [10,11,12,13,14] and their complex fabrication [12]. For example, Zhao et al. reviewed optical fiber-integrated MSs for applications, such as telecommunication, sensing, imaging, and biomedicine [10]. Holloway et al. focused on the development in recent years of such MSs from microwave to optical regimes [11]. Su et al. reviewed the fabrication and applications of MSs [12]. Glybovski et al. reviewed different types of MSs for a broad range of the operational wavelengths for wavefront shaping, lenses, and polarization transformation [13]. In this review paper, we focus on a special application of the MS, i.e., polarization conversion. This conversion using MS can be categorized into three main types: transmission-based, reflection-based and dual functional (both transmission and reflection). Figure 1 depicts the functionality of MS for polarization conversion. Figure 1a shows transmission-based polarization conversion, where incident linear polarization is converted into transmitted circular polarization. In this scenario, we termed MS as MS-T. Figure 1b shows the polarization conversion phenomenon using reflection-based MS. We termed this as MS-R. In Figure 1c, it is shown that MS performs polarization conversion for both transmitted and reflected waves. We termed such an MS as MS-T/R.

In this report, we describe advances in the field of transmissive MSs based polarization converters and their uses in shaping electromagnetic waves. Importantly, some key findings highlighting single band, wideband and multiband polarization conversion using MSs are discussed. Moreover, pattern selection for the desired operation has been discussed. Several controlling methods for polarization conversion are discussed and future directions for fundamental research and applications are identified. 

## 2. Why Polarization Conversion Is Required

Amplitude and polarization direction can characterize any electromagnetic wave. This polarization direction represents the direction of oscillations an electromagnetic wave carries while it passes through the medium. The polarization state of an electromagnetic can control electromagnetic waves in many ways, e.g., polarization rotation and polarization conversion, often termed polarization transformation. The term ‘polarization conversion’ means the conversion of one form of polarization into another (linear polarization (LP) to circular polarization (CP)). Polarization converters have enormous applications, depending upon the operating region they are being used. They are used from microwave to optical regions, depending upon the applications that need CP.

In microwave and millimeter-wave communications, the polarization selection of the antenna highly depends on the application and medium. In such a case, polarization mismatch, Faraday’s rotation by the ionosphere, and multipath fading significantly disturb and degrade the performance of the channel. Therefore, CP waves are mostly used for command guidance, global navigation systems (GNS), and satellite communication systems because of their better performance than LP waves for their lower sensitivity toward the multipath fading and Faraday’s rotation. CP waves can be generated or transformed using a polarization conversion phenomenon. The former has many disadvantages, e.g., the CP horn antenna is cascaded with a polarizer, such as a screw circular polarizer [15,16], a built-in dielectric-based polarizer which makes these antennas costly and complicated. Moreover, CP antennas, once designed and fabricated, can only perform as CP antennas. On the contrary, CP waves obtained from the conversion phenomenon allow antennas to perform as LP antennas as well. Moreover, the LP antennas are simpler to design.

The terahertz (THz) band, lying between 300 GHz to 3 THz of the electromagnetic spectrum, has been investigated in numerous applications, such as bio-imaging, spectroscopy, environmental surveillance, biochemical, and pharmaceutical sciences [17,18,19]. Circular polarized THz waves are used in biomedical imagining due to the different chirality of biomolecules for CP THz beams. Similarly, they are used in biosensing and drug delivery [17,20]. Therefore, LP-to-CP converters in THz technology, which are termed quarter-wave plates (QWPs), are very important for THz systems and advancements in their performance can have a significant impact on existing THz systems’ performance.

## 3. History and Modeling of MSs

Before further discussion, we clarify here that those planar 2D arrays whose periodicity is not smaller than operating wavelengths will not be called MSs here, because the term ‘MS’ is associated with cells whose periodicity is smaller than a wavelength. Here, the wavelength is a term used for λ_freespace_. Mesh and wire-based structures have been extensively explored for antenna systems to realize polarizers, which were based on averaged boundary conditions and, later, the first homogenization theory of artificial electromagnetic surfaces emerged. Frequency selective surfaces (FSS) are another type of planar array which exhibit resonant type transmission and reflection characteristics. Dipole and aperture based FSS have been employed for microwave filters operating with plane waves.

MSs were termed nano-islands in the starting period of their explorations in the optical region. Their modeling was carried out using oblate spheroids [21,22,23] to study their different characteristics, such as frequency and polarization selectivity. For most of the applications, the only electric response was not sufficient, therefore, an accompanying magnetic response was required to be considered in MS [24,25]. The homogenization model of MS presented by [26,27,28] does not cover bi-anisotropic MS. Such MSs have been targeted for polarization controlling applications for chiral [29,30] and omega type anisotropicity [31]. Chiral bi-anisotropic MSs for polarization conversions take advantage of inclusions which are polarized by electric or magnetic fields. For such MSs, simultaneous parallel electric or magnetic fields can exist. Another type is the omega type inclusions, where electric or magnetic dipoles are directly orthogonal to each other due to applied electric or magnetic fields only. These inclusions also help in achieving polarization conversion.

Unlike the reviews [11,12,13,14,15,16,17,18,19,20,21,22,23,24,25,26,27,28,29,30,31,32,33,34] which described several MSs for different applications, we aim to create a link between single and multiband MSs for polarization conversion applications in microwave and optical regimes. MS excitation can be modeled as an incident plane wave with wave numbers K_1,2_, and wave impedance η_1,2_. Modeling of MS can be carried out using two approaches, the surface susceptibility approach and the equivalent circuit approach. The former approach has a disadvantage, as in some bi-anisotropic MSs’ surface susceptibility has no physical meaning [14]. Whereas the latter provides an inside to physical procedures. Nonetheless, the common thing is both models characterize MS as a homogeneous sheet. In the surface susceptibility approach, MS can be represented by electric and dipole moments [35].
(1)$$\vec{P}=\varepsilon \overset{=}{\chi_{ee}}\vec{E_{av}}+\sqrt{\varepsilon \mu }\overset{=}{\chi_{em}}\vec{H_{av}}$$
(2)$$\vec{M}=\overset{=}{\chi_{mm}}\vec{H_{av}}+\sqrt{\varepsilon \mu }\overset{=}{\chi_{me}}\vec{E_{av}}$$where $\overset{=}{\chi_{ee}}$, $\overset{=}{\chi_{mm}}$, $\overset{=}{\chi_{em}}$ and $\overset{=}{\chi_{me}}$ are the electric/magnetic susceptibilities describing the response to electric/magnetic excitations, and ‘av’ denotes average fields on both sides of MSs. The above equations can be used to solve many boundary value problems for MSs. As discussed before, the solution for MSs under the susceptibility approach has some shortcomings, such as no physical insight in some cases.

Alternatively, MSs can be modeled using an equivalent circuit approach. Let us consider a plane wave striking MS either from medium 1 or medium 2, as shown in Figure 2. We can simulate/measure r_1_, r_2,_ and ‘t’, assuming that ‘t’ remains the same for an incident wave from medium 1 or medium 2 as MSs are reciprocal. Using the homogeneous sheet model of MSs described by [36], we can retrieve r_1_, r_2,_ and ‘t’. Using transmission line modeling of two-port passive and reciprocal devices, MS can be represented by an equivalent T-circuit with two series impedances Z_1_, Z_2,_ and a shunt admittance Y. Using r_1_, r_2,_ and ‘t’, Z_1_, Z_2,_ and Y can be calculated for the MS [14]. Detailed modeling work has already been carried out [37]. Modeling is out of the scope of this review, therefore we will not discuss its details.

For incident oblique waves, if MS has non-zero electromagnetic susceptibility, susceptibilities $\overset{=}{\chi_{ee}}$, $\overset{=}{\chi_{em}}$, and $\overset{=}{\chi_{mm}}$ are independent of the incident angles. In this case, three equivalent circuit parameters Z_1_, Z_2,_ and admittance Y can be represented in terms of scalar susceptibilities [14].

Metasheets are a class of MSs based on uni-layered substrate. They can be further classified into electric metasheets and magneto-electric metasheets. Electric metasheets are those metasheets which only have electric polarization when an incident electric field is applied. Such sheets are single metallic patterned over the substrate and/or have metal ground. They can be described through a few equivalent parameters. For example, a metasheet based on metal gratings can be described by grid impedance, whereas a metasheet based on planar array patches over the ground can be described as surface impedance [38]. Applied electric field on such sheets does not produce induced current loops inside them, being extremely thin layered dielectric sheets to act as an equivalent sheet of magnetic surface current. Examples are graphene sheets, single metallic layered sheets, such as dense wires and strip grids, etc. As the incident electric field produces only electric dipole moments, FSS can also be considered as a metasheet because FSS also has just an electric response, as they require having frequency-selective transmission and reflection. FSS can be categorized as the patterns which can be equalized as inductances, such as inductive thin wires or strips; capacitances, such as an array of patches, and the one which has both inductances and capacitive behavior.

If MSs have a non-zero but still negligible thickness as compared to the wavelength, electric current loops arise inside the MSs even if they are not closed. Usually, MS has both electric and magnetic currents. Induced electric currents on the MS are comprised of two parts: symmetric and asymmetric currents concerning the middle plane. The asymmetric electric current in each unit cell gives rise to magnetic surface currents. In spite of advanced functionalities in MSs, control over transmission and reflection properties still need to be explored. This needs the induction of independent electric and magnetic currents. In bi-anisotropic MSs, incident electric or magnetic polarization will provide induction of both electric and magnetic fields. Most generic relations for these induced electric and magnetic fields are [14]:(3)$$\left[\begin{array}{c} J_{e}\\ J_{m} \end{array}\right]=\left[\begin{array}{cc} Y_{ee} & Y_{em}\\ Y_{me} & Y_{mm} \end{array}\right].\left[\begin{array}{c} E_{inc}\\ H_{inc} \end{array}\right]$$
where $J_{e}$ and $J_{m}$ are equivalent electric and magnetic current densities, $Y_{ee},\;Y_{mm},\;Y_{em}$ and $Y_{me}$ are corresponding admittances to $\overset{=}{\chi_{ee}}$, $\overset{=}{\chi_{mm}}$, $\overset{=}{\chi_{em}}$ and $\overset{=}{\chi_{me}}$. As mentioned previously, of the three types of MSs, reflection-based MSs—also called impenetrable MS—are described by surface admittance. Therefore, to consider this, let us consider a case of patches (capacitive array) whose impedance is presented by $Z_{p}=\frac{1}{jwC}$. The wave penetrating from patches passes through the substrate and is reflected by the ground plate. The substrate acts as a small transmission line of thickness ‘d’ and wave impedance η_2_, whose impedance is $Z_{d}=jwL$. Thus, the surface impedance of this MS turns out to be Zd || Zp. Hence, this sheet can be characterized by an impedance of homogeneous MS. Here, ‘d’ behaves as an effective magnetic response which induces magnetic surface currents [39]. This concept is extended in Section 6.1, where it presents the concept of polarization conversion, though in transmission mode, where the ground plate will not be present to allow transmission.

Frequency-selective filtering is a widely used operation in the microwave and optical regions of communication systems. FSS is used in this regard due to its very good transmission characteristics. They have a wide range of applications, such as splitters, duplexers, and radomes. They can achieve bandpass and bandstop properties depending upon the size, type, and shape of patches and substrate material. A detailed description of the calculations of FSS behavior can be found in [40,41]. Nonetheless, FSS requires the unit cell to be at least half a wavelength, although it may operate at multi-resonances as well. However, if inclusion dimensions are changed to the subwavelength scale in such a way that inclusions still have resonant behavior, MSs can replace FSS [42]. It has several advantages, such as miniaturization, and lower sensitivity to the plane wave’s angle of incidence. Another major attraction is for limited space, where more units fit the area [43], which is very useful for applications that have limited space, e.g., radomes. Complex shaped frequency-selective MSs can be designed for required applications by using independent inductive and capacitive elements, e.g., refs. [44,45] used this concept by introducing lumped on-chip elements realizable in the microwave region. Another approach is to use patches for neighboring capacitive and inductive cells [46]. It is important to mention that, at these resonances, the structure provides very high electric polarizability. For the multiband operation of such frequency-selective MSs, one possible way is to use two complementary structures with the same periodicity on each side of the MS so that two parallel resonances can exist [47]. Another approach is to use the same geometry and shape different sizes according to the required multiple frequencies both present on a unit cell [45]. This concept can be applied to multiband polarization converters as well.

Having discussed simpler cases of achieving transmission curves/reflection curves, we move to the requirement of different phase responses or, in other terms, phase control of microwave and optical waves. To have such a response, one needs to have a magnetic response in addition to the electric response. In this case, when an incident wave strikes MS from one side, the structure can be designed for desired operations, such as phase-shifting and optical activity.

## 4. Polarization Manipulation Using MSs

As discussed in Section 2, the polarization of electromagnetic waves possesses a very important role in radio and optical communication systems. They have been designed and explored for applications, such as dual-polarized radars, fiber optic communications, and MIMO systems. These devices are called polarization manipulators. Their most important type of polarization is a linear polarization wave which can be rotated from one polarization direction to another, converted from one state to another, or selected depending upon their polarization state for reflection or transmission. 

Polarization rotators are also termed polarization twisters or twist-polarizers. Initially, they were used for satellite ground stations in dual-polarized antennas [48]. In optoelectronics applications, they are used for displays [49]. Optical activity of natural materials as observed in the Nicol prism, the lenses of polarized sunglasses and proteins are found to be weaker [49], which means rotation of polarization plane per wavelength is low. Therefore MSs are advantageous in this manner. Polarization rotators were designed for the first time using dense wire grids with a multilayer structure whose wire rotates from one layer to another [48,50]. They can be designed in such a way that they miniaturize the return losses. However, such devices only work for a particular linear polarization. Chirality in the optical regime was reported in [51]; the device shows the rotation of 250°/λ. Similarly, another polarization rotator was reported [52] which had only λ/30 thickness and consisted of a complementary split-ring resonator (SRR). MSs can also be employed to possibly engineer the sensitivity to polarization states, such as linear or circular. Devices in line with such properties to improve the properties of microwave antennas were proposed [53,54]. These devices control circular polarization, as they allow for one form of circularly polarized waves, termed circularly polarized selective surfaces (CPSS). These surfaces are further categorized as symmetric and asymmetric CPSSs where the symmetric allows one type of CP wave to transmit through them, irrespective of the direction.

Polarization manipulation has also been explored for non-contact Hall measurements by the magneto-optic effect, where [55,56] study the chiral molecular structure of proteins and DNA [57,58]. One of the most important functions of MSs under polarization manipulation is linear polarization to circular polarization conversion (LP-to-CP conversion) in transmission mode. In addition to many applications of LP-to-CP converters in optical regimes [59], these converters are being used in microwave and millimeter-wave systems, including imaging systems and reflectors [60,61].

## 5. Theory of Polarization Conversion

The basic concept for polarization converters is to use a combination of capacitive and inductive guides. Several such structures have been realized through anisotropic metal sheets [62,63,64]. MS-based polarization converters can be used in line with this concept, i.e., transmitted co-polarized and cross-polarized components should have the same amplitude, however, should be 90° phase distinct. MSs consisting of the meandered pattern are designed in such a way that they behave as inductance if the applied tangential electric field is parallel to the strips and as capacitances if the electric field is perpendicular to the strips.

For the detailed operation of the LP-to-CP converter, let us consider a plane incident electric field $\vec{E_{xi}}$ and $\vec{E_{yi}}$ with x-polarization and y-polarizations, respectively. Ignoring the reflection losses and considering MS as an ideal transmissive polarization converter, $\vec{E_{xi}}$ and $\vec{E_{yi}}$ waves traveling in ‘+z’ are applied on MS. Transmitted waves $\vec{E_{xo}}$ and $\vec{E_{yo}}$ can be expressed in terms of transmission coefficients, as shown in Equation (4). Here, ‘xo’ and ‘yo’ represent outgoing waves corresponding to the incident x-polarized and y-polarized waves, respectively. These transmitted waves can be represented in terms of the transmission matrix ‘T’ for $\vec{E_{xo}}$ and $\vec{E_{yo}}$, as shown in Equation (4).
(4)$$\left[\begin{array}{c} \vec{E_{xo}}\\ \vec{E_{yo}} \end{array}\right]=T\left[\begin{array}{c} \vec{E_{xi}}\\ \vec{E_{yi}} \end{array}\right]=\left[\begin{array}{cc} \vec{t_{xx}} & \vec{t_{xy}}\\ \vec{t_{yx}} & \vec{t_{yy}} \end{array}\right]\left[\begin{array}{c} \vec{E_{xi}}\\ \vec{E_{yi}} \end{array}\right]$$
(5)$$\mathrm{Where},\;\vec{E_{xi}}=E_{xi}\vec{e_{x}}=E_{o}e^{-jKz}\vec{e_{x}}$$

Combining magnitude and phase responses of Equations (4) and (5):(6)$$\left[\begin{array}{c} \vec{E_{xo}}\\ \vec{E_{yo}} \end{array}\right]=\left[\begin{array}{cc} \left|t_{xx}\right|e^{j\emptyset_{xx}} & \left|t_{xy}\right|e^{j\emptyset_{xy}}\\ \left|t_{yx}\right|e^{j\emptyset_{yx}} & \left|t_{yy}\right|e^{j\emptyset_{yy}} \end{array}\right]\left[\begin{array}{c} \vec{E_{xi}}\\ \vec{E_{yi}} \end{array}\right]=\left[\begin{array}{cc} \left|t_{xx}\right|e^{j\emptyset_{xx}} & \left|t_{xy}\right|e^{j\emptyset_{xy}}\\ \left|t_{yx}\right|e^{j\emptyset_{yx}} & \left|t_{yy}\right|e^{j\emptyset_{yy}} \end{array}\right]\left[\begin{array}{c} E_{o}e^{-jKz}\vec{e_{x}}\\ E_{o}e^{-jKz}\vec{e_{y}} \end{array}\right]$$
where, $t_{xx}$ and $t_{xy}$ are transmission coefficients for co and cross-polarization components, respectively, for incident x-polarized wave; $\emptyset_{xx},\;\emptyset_{xy}$ are the corresponding phases. Similarly, $t_{yx},\emptyset_{yx}$ and $t_{yy},\emptyset_{yy}$ are the transmission coefficient and phase for the transmitted x- and y-polarized waves for the incident y-polarized waves, respectively. $\left|t_{xx}\right|,\left|t_{xy}\right|$, $\left|t_{yx}\right|$, $\left|t_{yy}\right|$ can be computed as shown in Equation (7):(7)$$\left|t_{xx}\right|=\frac{\left|E_{xo}\right|}{\left|E_{xi}\right|},\;\left|t_{xy}\right|=\frac{\left|E_{yo}\right|}{\left|E_{xi}\right|},\left|t_{yx}\right|=\frac{\left|E_{xo}\right|}{\left|E_{yi}\right|},\;\left|t_{yy}\right|=\frac{\left|E_{yo}\right|}{\left|E_{yi}\right|}$$

Here, we are considering that the incident wave is an x-polarized wave. Transmitted waves will have two orthogonal parts: one in phase with the incident wave ($t_{xx},\emptyset_{xx})\;$ and the other is orthogonal to it, also called the cross-polarization component here ($t_{xy},\emptyset_{xy})$. Magnitudes for these transmitted waves will vary over frequency ranges. However, in the band of interest, if these transmission coefficients become comparable in magnitude and ±90° apart in their phases, i.e., $\left|t_{xx}\right|$≈$\left|t_{xy}\right|$ and $\emptyset_{d}=\emptyset_{xy}-\emptyset_{xx}=2n\pi \pm \pi /2$; where n=0,±1, ±2… is an integer. A transmitted wave will be called a circularly polarized wave. Transmission conversion performance for the transmitted wave in the microwave and milliemeter region is usually described by the axial ratio (**AR**) which can be defined in Equation (8) as [65]:(8)$$AR={\left(\frac{|t_{xx}{|}^{2}+{\left|t_{xy}\right|}^{2}+\sqrt{a}}{|t_{xx}{|}^{2}+{\left|t_{xy}\right|}^{2}-\sqrt{a}}\right)}^{1/2}$$
where ‘***a***’ can be calculated from Equation (9) as [65]:(9)$$a=|t_{xx}{|}^{4}+{\left|t_{xy}\right|}^{4}+2{\left|t_{xx}\right|}^{2}{\left|t_{xy}\right|}^{2}cos\left(2\emptyset_{d}\right)$$

As mentioned previously for ideal polarization conversion, $\left|t_{xx}\right|$ = $\left|t_{xy}\right|$ and $\emptyset_{d}=\emptyset_{xx}-\emptyset_{xy}=2n\pi \pm \pi /2$. In this case, the outgoing wave will be perfect circularly polarized and **AR** will be 1 (0 dB). Most of the systems allow **AR** in the range of 0~3 dB. However, a higher value of **AR** indicated that the transmitted wave will be slightly elliptical.

For THz and the optical region, the performance of transmitted polarized wave is calculated using stokes parameters, as in Equations (10)–(13) [66]:(10)$$I={\left|t_{xx}\right|}^{2}+{\left|t_{xy}\right|}^{2}$$
(11)$$Q={\left|t_{xx}\right|}^{2}-{\left|t_{xy}\right|}^{2}\;$$
(12)$$U=2*\left|t_{xx}\right|\left|t_{xy}\right|cos\left(\emptyset_{d}\right)$$
(13)$$V=2*\left|t_{xx}\right|\left|t_{xy}\right|sin\left(\emptyset_{d}\right)$$

As compared to the axial ratio in microwave and millimeter bands, in THz and optics regimes, ellipticity depicts the polarization of the outgoing wave, calculated as Ellipticity=V/I. Ideally, ellipticity as ‘−1′ and ‘+1′ depicts that the transmitted wave is an RHCP wave and LHCP wave, respectively. However, a value close to ±1 is considered to be acceptable. This analysis is essentially the same as the **AR** analysis for perfect circular polarization conversion. It is essentially just two different terminologies for microwaves and light waves.

Traditionally, the design of a polarizer (LP-to-CP converter) involves the equivalent circuits modeling of patterned MSs in combination with electromagnetic simulations. Values of equivalent circuit elements are obtained that fulfill the design conditions for co- and cross-polarizations, whereas EM simulations are used to find the corresponding values of circuit parameters. Of course, this is valid for polarizer designs in the microwave regime only. Another design approach comes into play when the circuit model does not allow standard design procedures. This design involves optimization which is applied to equivalent circuits. Most of the designs consider an incident wave tilted at 45° concerning cell periodicity. In this way, two independent models for each polarization are considered. Thus, the transmissive polarizer design involves the synthesis process of two different MS-based filters, while the required phase shift between them should be 90°. Therefore, most of the designs for MS-based LP-to-CP converters involve dual diagonal symmetric structures. Thus, such structures make it possible to describe the unit cell for two incident linear polarizations. For the equivalent circuit approach, the cell is analyzed with a new unit cell that exhibits two symmetries in both the horizontal and vertical planes, thus, their electrical responses at normal incidence do not generate vertical components. Hence, they are decoupled and can be treated as independent.

## 6. Review of Recent Progress of MSs Based Polarization Conversion

Reflection-based polarization converters are out of the scope of this review paper. Here, we present advancements in transmission-based linear to circular polarization converters using MSs from microwave to optical regimes. Transmission properties of MSs are categorized into amplitude control and phase control. Phase control methods are adopted for polarization rotation and conversion properties. Figure 3 shows the schematic flowchart for the review of MSs, specifically focusing only on transmission-based linear to circular polarization conversion.

### 6.1. Single Band Transmissive MS-Based Converters

MS was firstly used by Zhu et al. in 2013 [67] as a single-band polarizer. They proposed a singly layered structure whose each unit cell comprises a rectangular loop and a diagonal microstrip, as shown in Figure 4a. With the help of a diagonal strip, the pattern on the MS realized a diagonal symmetric structure. They could achieve around 9% bandwidth for a 3dB operation of the axial ratio. However, the performance of the polarizer for oblique incidences was not good enough as the polarizer remained stable for only 10° oblique incidences. One year later, Martinez-Lopez et al. [68] proposed four-layered split rings bisected by a metal strip based structure, as shown in Figure 4b, to realize LP-to-CP conversion. They used ring resonators instead of the square resonators as used by Zhu et al. because of the good reflection properties at the resonant frequency. They used a slant incident wave (tilted to 45°) to the x-axis so that if the incident wave could be decomposed into two orthogonal components, they could face different structures along the *x* and *y* axes. Since the structure was based on four layers, the fabrication of the design was complex, where they used acrylic glass frames as spacers to ensure the required distance. Metal dowel pins were used to align the elements of the layers. Due to the multiple layered structures, the bandwidth was improved to about 31%. Since the structure was based on a circular ring, the polarizer was relatively stable over 25° oblique incidences.

In later years, significant advancements in operational bandwidth were reported using simpler structures [69,70]. For example, Baena et al. proposed an LP-to-CP converter based on self-complementary zigzag MSs [70]. The structure consists of a bi-layered surface. The unit cell of the structure consisted of a zigzag metallic pattern, which acts as equivalent capacitances for incident horizontal polarization, whereas, for incident vertical polarization, it acts as equivalent inductive units. Such a structure proved to achieve very wide operational bandwidths for numerical simulations (70%); however, experimentally, they could achieve around a 40% axial ratio bandwidth.

Several research groups reported wideband polarization converters using square rings enclosed in different patterned structures [73,74,75,76]. This is due to the interesting property of the square ring as a wideband filtering structure without polarization conversion [77]. Most of the work reported used square ring based unit cells to achieve wideband polarization conversion performance. Transmissive MS-based polarizers (called quarter waveplates) have been reported in the optical regime as well. They used Babinet-inverted or silicon MSs based on Fano-resonances. Babinet-inverted MSs have been an attractive research field because of their ultra-thin properties and maximum transmission efficiency at resonance [78]. Wang et al. reported polarizers based on cross-shaped metallic cross apertures with unequal lengths to support two orthogonal resonant modes in the THz range [71]. By selecting proper suitable lengths of the apertures, equal transmission amplitudes, and 90° phase differences were observed. Table 1 enlists the key findings for a few MS-based LP-to-CP converters in transmission mode.

In order to describe polarization conversion using equivalent circuit theory, ref. [81] introduced a new unit cell, as shown in Figure 5a,b. It can be seen that the unit cell in Figure 5b can be seen as that in Figure 5c once the diagonal strips are removed. Incident wave ‘E’ can be divided into two orthogonal components as E1 and E2, as shown in Figure 5b,c. The equivalent circuit theory can be applied by superposition, using diagonal strips removed and with diagonal strips. The structure in Figure 5c is symmetric in both planes. With the diagonal strips along E2, equivalent inductances will appear and will produce a phase difference between two orthogonal components. The length and width of the strip can be optimized to achieve a 90° phase difference between the two orthogonal transmitted waves, resulting in transmitted circular polarized waves.

### 6.2. Multiband Transmissive MS-Based Polarization Converters

Multiband operation of communication systems to merge multiple systems into a single one has resulted in a reduction in size, cost, and complexities. A multiband circular polarization operation is a challenging task. A way to meet the challenge is through multiband circularly polarized (CP) antennas, such as multi-feed, helical, spiral, and shared aperture antennas [82,83,84]. Multiband CP waves can alternatively be obtained using multiband LP-to-CP converters. This solution is particularly attractive at high frequencies because of the complex feeding structures associated with CP antennas. There has been an increasing trend towards multi-functional polarization conversion devices [85,86] in the microwave regime. In this way, multiple systems can be merged to miniaturize the systems. However, multiband LP-to-CP converters generally operate over a narrow range of frequencies and have low stabilities over oblique incidences.

The design principle for multiband and multi-functional polarization converters is essentially the same as those of the single-band polarization converters, i.e., a given LP incident wave, when transmitted through an MS with a 90° phase difference between two orthogonal components, converts the incident linear wave into a CP wave, however, this time, in multiple bands [65]. Either a systematic equivalent circuit design approach has been proposed to design such multiband converters [65,87] or a step-by-step design guideline in an EM simulator can be followed [88,89,90,91,92]. For the multiband operation of such converters, one possible way is to use multiple resonant structures with the same periodicity on each side of the MS so that two multiple resonances can exist. Another approach is to use the same geometry and shape different sizes according to the required multiple frequencies both present on a unit cell. This concept was used in multiband converters. Only asymmetric elements based unit cells along the x- and y-axes can result in the transmission of orthogonal components. Therefore, an anisotropic MS can be the right choice for such converters. The split rings have been proven to be a very good candidate for such multiband converters [65,87,89,91,92,93]. These splits react differently to two orthogonal LP fields and their mutual couplings between their ends play an important role in achieving opposite handedness for the transmitted CP waves. Either a design operation using the equivalent element extraction can be followed [65,87,93], or step-by-step design guidelines using an EM simulator can be used to design such converters [88,89,94].

This design technique for multi-resonant structures was used by Zeng et al. in [86] to obtain dual-band polarization conversion operation. The structure is a four-metallic layered structure and consists of a cascaded structure of split-ring resonators and metallic patches divided by a metallic strip. The amount of 31% and 13% bandwidths were obtained for the two bands of operation under the provision of 25° oblique incidences. A similar concept was used by Naseri et al., however, the structure was a tri-layered structure and the bandwidth of operation was just 2.5% and 1.7% [88,93]. The concept of multi-resonant structures within the same periodicity was used by Wang et al. using Jerusalem-cross MSs to obtain a dual-band LP-to-CP converter in Ku- and Ka-bands [85]. They could achieve 13% and 10% bandwidths for both bands of operation. Figure 6 shows the different multiband polarization converters with experimental setups.

It is important to mention here, however, that MS-based LC-to-PC converters in the microwave and THz regimes have different fabrication processes, characterization, and applications. Nevertheless, the design principle remains the same, i.e., an anisotropic structure resulting in co- and cross-polarized components having the same transmission value but a phase difference of 90°. Table 2 enlists the key findings for the state-of-the-art transmission-based multiband LP-to-CP converters. Split circular ring-based structures were used by [89,91,92], where either multiple-sized rings were used for multiple operations [81,91,92] or the concept of a cascading band reject and bandpass filter to realize multiband operation was used [89]. Table 2 enlists the key summarized findings for the state-of-the-art multiband transmissive polarization converters.

For the characterization and measurements of these multiple band LP-to-CP converters, usually, multiple antennas are used due to the unavailability of a single wideband antenna covering both/all operating bands. The distance between the antennas and the polarizer should satisfy the far-field condition, which is given by the following, to obtain an accurate measurement result: where $d_{f}\geq 2\frac{D^{2}}{\lambda }$, *D* is the largest dimension of the antenna, and *λ* is the minimum free-space wavelength. Most of the polarizers are characterized in a far-field region [65,81,91]. However, these multiband polarizers can be placed in the near radiative field as well [89,92].

### 6.3. Reconfigurable Transmissive MS Based Converters

Polarization conversion from MSs can be achieved by designing a pattern providing a certain distribution of electric and magnetic fields. Therefore, these operations are highly dependent on the pattern/structure of the unit cell, dielectric substrate and metallic layers material. Once fabricated, it becomes difficult to adjust the resonance frequency or properties. Single-band MS operations for LP-to-CP conversions present limitations in terms of the bandwidth and frequency of operation. This is because the intrinsic properties of resonating cells (LC resonators) vary as the frequency and bandwidth of operations are changed. For multiple frequency band operations, controllable MSs become essential. This not only helps in the reconfigurability of MS-based LP-to-CP converters but also can integrate multiple functionalities into a single MS. Tunability can be introduced by some external stimulus control mechanisms, such as controllable circuit elements, state-changing and structural changing.

Control through controllable circuit elements mostly requires varactor diodes [95,96,97]. Tuning methods using structural change and state change can cover microwave to optical regions. However, the third method is only applicable to the microwave region. State change control can be employed by using substrates, such as ferrites [98], semiconductors [99] or phase changing materials [100]. Structural change can exist by using microfluidic [101], gravitational field [102] or microelectromechanical (MEMS) systems [103]. Next, we will discuss each controlling method one by one.

#### 6.3.1. Controllability through Circuit Elements

The tunability of MSs was experimentally demonstrated with meta-atoms controlled by external knobs involving varactor or pin diodes in these meta-atoms. Each unit can be controlled by applying an appropriate voltage to the diode.

There have been interesting controlling circuit elements for the MS-based polarization converters. For example, a switchable LP-to-CP converter based on MS was proposed by Li et al., where controllability can be achieved by changing the states of double PIN diodes [104]. Incident LP waves can be converted into CP waves whose handedness is determined by the state of PIN diodes waves. Diagonal asymmetry in the unit cell of the polarizer is loaded with PIN diodes (SMP 1345-079LF). Figure 7a shows the unit cell for the proposed polarization converter. Top and bottom rings are fabricated on each layer so that biasing by DC power can be done easily. The orientation of the slots of top and bottom rings is varied (vertical and horizontal, respectively). The biasing of PIN diodes is done by pairs of DC lines shown on the left of Figure 7a. Incident LP waves are converted to CP waves when the PIN diodes of the top ring and bottom rings are in different states. A similar concept was used by Ma et al., who converted the incident LP wave into the transmitted CP wave by using a double-layered active MS. The structure works by tuning the working states of the PIN diodes in its unit cells [105]. Figure 7b shows the unit cell of the polarizer. Figure 7c–e show three states of the polarizer. In state 1, as shown in Figure 7c, diodes 1 and 3 are switched on, while diodes 2 and 4 are off. In state 2, diodes 1 and 3 are shut off while diodes 2 and 4 are switched on, which is shown in Figure 7d. Lastly, in state 3, as shown in Figure 7e, all the four diodes are switched on, leading to cross-polarization conversion.

A similar design consisting of active MSs based on voltage-controlled varactor diodes was proposed by Tao et al. [106], where the transmission response from the MS was tailored by the varactor diode; it behaved as a multi-functional polarization conversion. When an LP wave is made incident on the MS, it becomes converted to a CP one. Dual-band operation is realized in a single voltage configuration, resulting in transmitted orthogonal CP waves within two bands of operation. Moreover, in a single voltage configuration, two circular polarized waves with different handedness can be obtained at different frequencies by varying the applied voltage along the MS. When PIN diodes are on, MS operates in reflection mode and reflects the incident wave with the same polarization. Thus, different circuit elements can be used to achieve controllability in polarization conversion and multi-functional operations.

#### 6.3.2. Controllability through State Change

Another mechanism to control MSs for their LP-to-CP conversion is based on different materials, such as graphene. This is a 2D layered material based on carbon atoms arranged in a honeycomb lattice and has excellent electrical properties, which can help in controlling the interaction between light and matter. Graphene layers placed on the top of the insulator have proved to exhibit strong plasmonic behavior [107]. Guo et al. proposed a graphene-based LP-to-CP converter in transmission mode [108]. The phase difference between transmitted orthogonal waves can be controlled by varying the Fermi level of the graphene. Starting from conceptual diagram for controlling the polarization state of the outgoing wave using the Fermi energy level, the unit cell is designed and the phase difference between transmitted co- and cross-polarization components is controlled by varying Fermi energy levels. Similarly, Zhang et al. used graphene to dynamically regulate the operating band in the THz regime by controlling the Fermi energy of the graphene sheets [109]. The polarizer performs over 2.64–3.29 THz (21.92%) in the case of Ef = 0.1 eV. Figure 8a shows the periodic array configuration and corresponding controllability in the band of operation in Figure 8b.

A similar scheme is applied to the far-infrared region, i.e., the phase retardation between two orthogonally polarized wave components is constructed through light propagation. Chen et al. implemented an LP-to-CP converter based on ultra-thin MS to operate in the mid-infrared region (Figure 9a) [110]. Simulated results for transmission are shown in Figure 9b. It is clear that in the wavelength 2729 ~ 3094 nm, the phase difference of 90° between transmitted orthogonal waves is achieved. Figure 9c shows the SEM image for the fabricated sample.

In another exemplary work, Wang et al. proposed a switchable ultra-thin terahertz LP-to-CP converter by hybridizing a phase change material, vanadium dioxide (VO2), with an MS, as shown in Figure 9d. Meta transition for VO2 changes the operational frequency from 0.468 THz to 0.502 THz. MS performs transmissive linear to circular polarization conversion at two stated operational frequencies. They fabricated MS using pulsed laser deposition (PLD) on a c-cut sapphire substrate using a metallic vanadium target in an oxidizing background. Further, iron milling was used to pattern cross-shaped resonators. Figure 9e shows a resistive heater with a square aperture, which was milled at the center to vary the temperature controlled by an external voltage source.

#### 6.3.3. Controllability Using Structural Change

Several other controlling factors for LP-to-CP converters could be microfluidic MSs [112], as shown in Figure 10a; where, three different polarization states (LHCP, RHCP, LP) are obtained by microfluidic injection through different channels. Such a type of control is very useful for applications requiring a large number of diodes as tunable or MSs [113]. As seen in Figure 10a, when microfluidic is injected into the upper channel only, the outgoing wave is LHCP and vice versa is done for RHCP.

Some new controlling parameters could be gravity, as achieved by [102], and MEMS [114]; however, these controlling options have not been explored yet in transmissive MS-based LP-to-CP converters.

## 7. Conclusions

In this paper, MSs are classified based on their operational modes, where their transmission mode for polarization conversion applications is discussed in detail. A brief history of MSs and advancements in their modeling approach is presented. Further, detailed theories for polarization conversion and design strategies are discussed.

The authors have only focused on transmissive MSs for polarization conversion. To review all the functionalities of transmissive MSs, or to discuss both transmissive and reflective MSs is possible but hardly constructive. Our observation of the trends of transmissive MSs for LP-to-CP converters can be concluded as the first trend of miniaturization and easy-to-fabricate MSs. The second trend is a multiband operation to facilitate the merging of multiple systems together. Design ideas from microwave MSs have been transferred to optical MSs but less frequent, and vice versa. The third trend is the controllability of MSs using structural change, and the state change of controllable electric components. It is hoped that the present review will help to establish a link between the advancements in MSs and polarization conversion performance in microwave and optical regimes. The authors also suggest new possibilities to control/tailor polarization conversion performance using gravity [102] and MEMS [114], as these fields have not been explored for transmission mode LP-to-CP converters.

## Figures and Tables

**Figure 1 nanomaterials-12-01705-f001:**
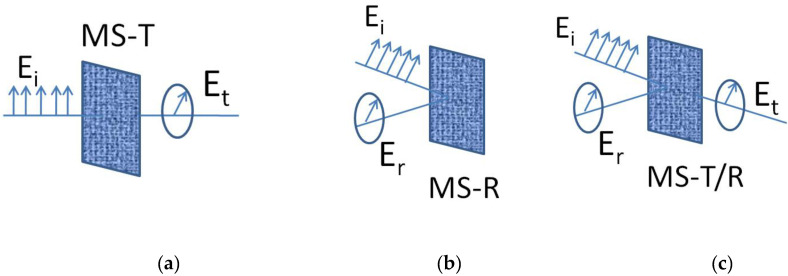
Selected polarization conversion functionalities of MS: (**a**) Transmissive polarization conversion (**b**) Reflective polarization conversion (**c**) Transmissive and reflective polarization conversion.

**Figure 2 nanomaterials-12-01705-f002:**
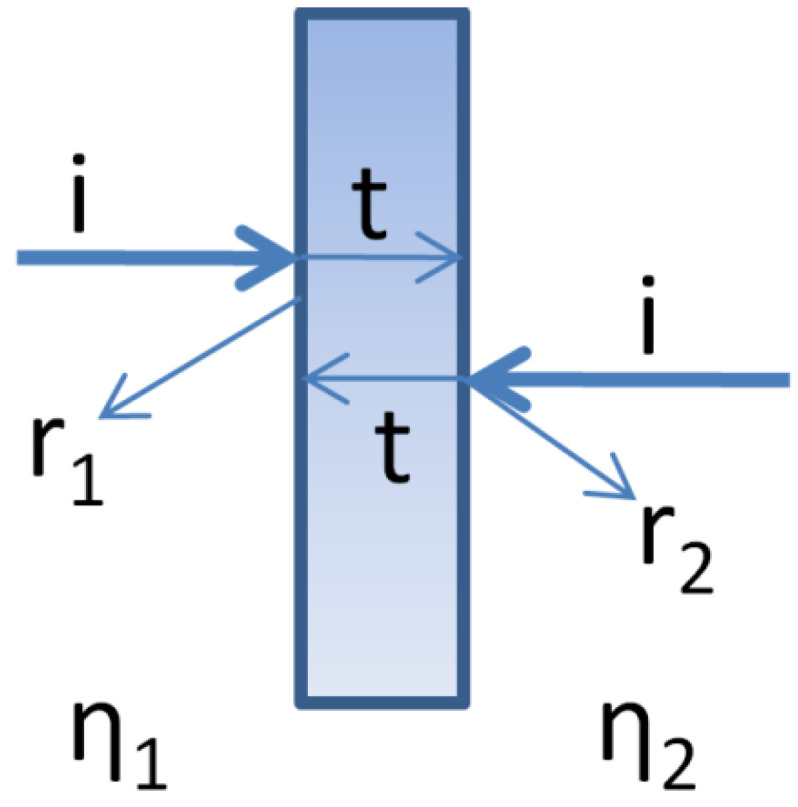
MS structured as an interface between two homogeneous isotropic media.

**Figure 3 nanomaterials-12-01705-f003:**
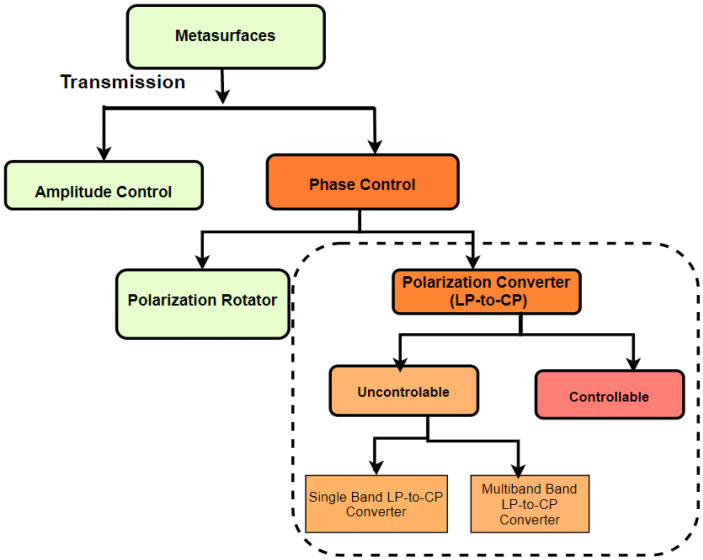
Scheme for the article.

**Figure 4 nanomaterials-12-01705-f004:**
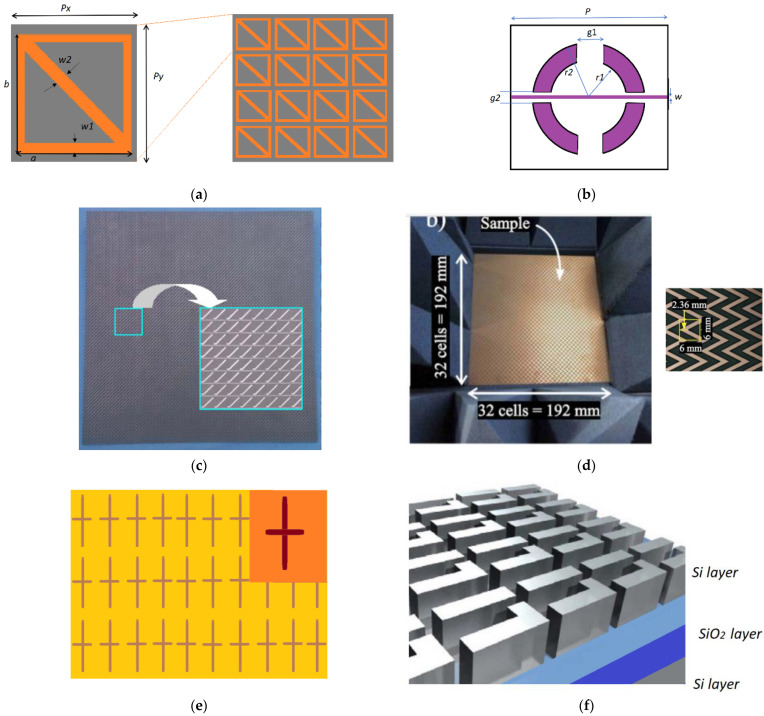
Different multi-layered single-band MSs (**a**) for linear to RHCP converter [67], (**b**) four-layered linear to circular polarization converter [68], (**c**) bi-layered polarization converter [69], (**d**) bi-layered zigzag structure, Reprinted, with permission [70], (**e**) bi-layered Babinet-inverted MSs [71], (**f**) Fano-resonant Si-MSs [72].

**Figure 5 nanomaterials-12-01705-f005:**
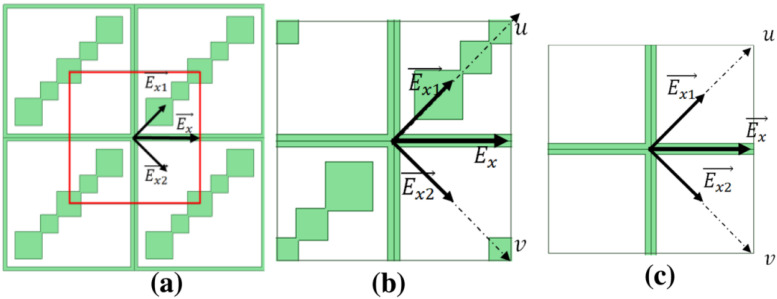
Equivalent circuit approach for polarization conversion operation (**a**)array for MS-based LP-to-CP conversion [81], (**b**) new unit cell [81] and (**c**) new unit cell with diagonal strips removed [81].

**Figure 6 nanomaterials-12-01705-f006:**
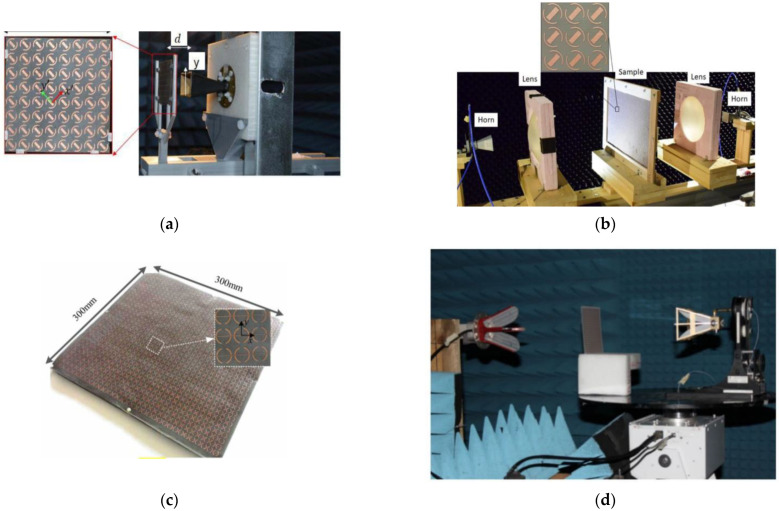
Multiband polarization converters; (**a**) 3D printed setup to hold the horn and polarizer (Reprinted, with permission, [88]; (**b**) quasi-optical measurement setup for dual-band converter (Reprinted, with permission, [93]; (**c**) sample sheet for dual-band LP-to-converter [84]; (**d**) free-space measurement system used in [81].

**Figure 7 nanomaterials-12-01705-f007:**
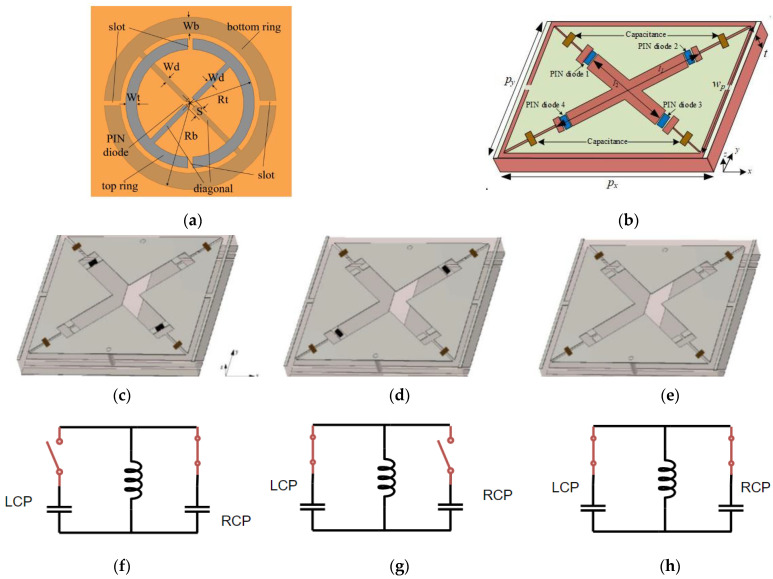
(**a**) DC circuit diagram for polarization control and unit cell of the polarizer (Reprinted, with permission [104]); (**b**) unit cell of the active MS with side view of the top layer and dielectric substrate (**c**) state 1 (**d**) state 2 (**e**) state 3. (**f**) Equivalent scheme for state 1; (**g**) equivalent scheme for state 2 (**h**) Equivalent scheme for state 3 [105].

**Figure 8 nanomaterials-12-01705-f008:**
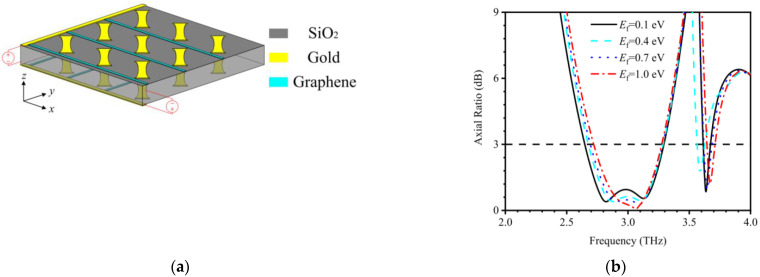
Schematic of an ultra-thin quarter-wave plate MS made of an array graphene and gold (**a**) periodic array configuration [109]. (**b**) The **AR** curves when Ef runs from 0.1 eV to 1.0 eV [109].

**Figure 9 nanomaterials-12-01705-f009:**
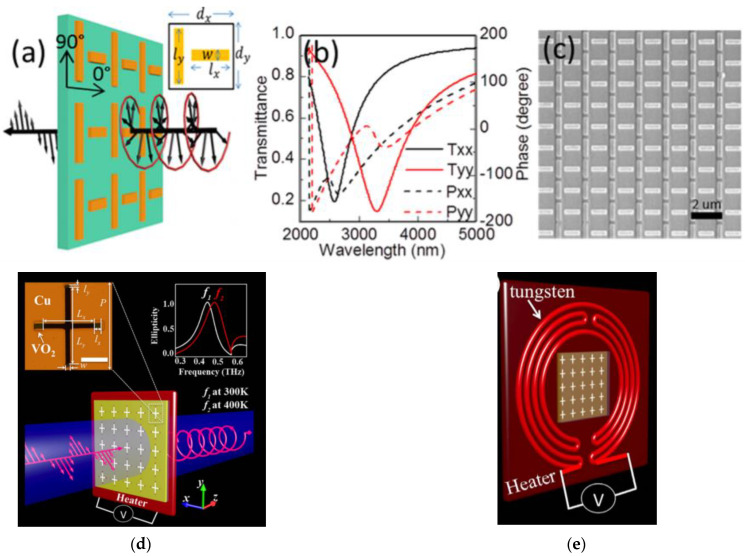
(**a**) Schematic of the design of MS as polarization converter in mid-infrared region. (**b**) Full-wave simulations of the transmittance (solid curve) and phase (dashed curve) for the MS; (**c**) SEM image of a nanoimprinted MS [110]; (**d**) experimental schematic for frequency switchable of the THz LP-to-CP converter; (**e**) schematic backside view of the resistive heater with a square aperture (6 × 6 mm^2^) milled at the center to allow THz to pass through [111].

**Figure 10 nanomaterials-12-01705-f010:**
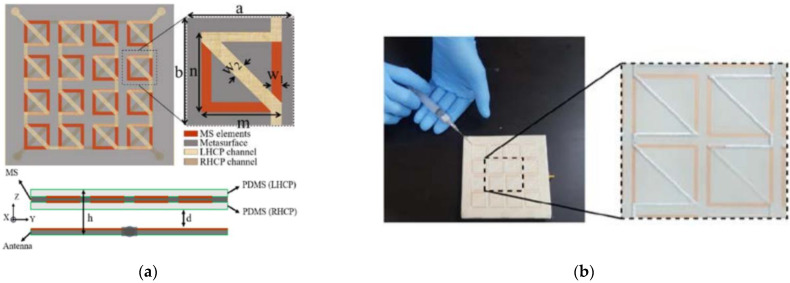
(**a**) MS-based polarizer with microfluidic channels (Reprinted, with permission [112]); (**b**) fabricated prototype (Reprinted, with permission [112]).

**Table 1 nanomaterials-12-01705-t001:** State-of-the-art single-band MS-T-based LP-to-CP converters.

Ref	Operating Frequency	Operating Bandwidth	No. of Metallic Layers	Angular Stability
[67]	2.45 GHz	9%	1	10
[68]	31	34%	4	25
[69]	14.05	60%	2	-
[70]	9	70% (numerical), 40% (experimental)	2	25
[79]	3	5%	1	-
[73]	15.3	40%	3	-
[74]	9.38	37.3%	2	75°
[80]	8.9	11%	2	±30°
[75]	29.5	7.4%	3	-
[76]	1205	43.9%	2	-
[71]	870	-	2	-
[72]	λ = 4.55 μm	-	2	-

**Table 2 nanomaterials-12-01705-t002:** State-of-the-art transmission-based multiband MS-based LP-to-CP converters.

Ref	Operating Frequencies	Operating Bandwidth	No. of Metallic Layers	Angular Stability	Orthogonality
[65]	7.6, 13	31.6, 13.8	4	25	same
[87]	18.95, 28.5	13.1, 10.5	2	20	orth
[88,93]	19.95, 29.75	2.5, 1.7	3	30	orth
[89]	17.8, 36.5	25, 16.4	2	-	orth
[91]	20.6, 29.71	5.56, 3.97	2		orth
[92]	9.5, 12.5	4.2,6	3	55	orth
[94]	0.73, 1.33	24%, 30%	1	-	
[81]	8.45, 28.4, 38.8	27.22, 21.2, 17.5	2	25	orth

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
