# Peer review of "Transmissive Polarizer Metasurfaces: From Microwave to Optical Regimes"

_nanomaterials, 2022, doi:10.3390/nano12101705_

Round 1

Reviewer 1 Report

This paper proposes a comprehensive approach about the relevance of metasurfaces for conversion from linear polarization to circular polarization.

The general organization of the paper is very clearly presented. The concept of metasurfaces is correctly introduced, and their modelling is correctly presented. Figures clearly detail the geometrical configurations that are considered and the parameters investigated. The bibliography is extensive, reporting on a synthetic manner all the relevant results already published, for already established structures as well as for advanced 2D material structures.

This paper is a very good benchmark for the topic.

Author Response

Dear Editorial and Production Board, 

Nanomaterials,

We thank this valuable reviewer for the critical feedback. We feel lucky that our manuscript went to this reviewer as the valuable comments not only helped us to improve the manuscript but suggested some neat ideas for future studies. Please do forward our heartfelt thanks to these experts.  Based on the received comments, careful modifications have been made to the manuscript. All the changes have been made after track changes and are highlighted in yellow color in the manuscript. We hope our revised manuscript will meet the standard of Nanomaterials. Below, you will find point-by-point responses to the first reviewer’s comments and suggestions.

Response to Reviewer 1:

This paper proposes a comprehensive approach about the relevance of metasurfaces for conversion from linear polarization to circular polarization.

The general organization of the paper is very clearly presented. The concept of metasurfaces is correctly introduced, and their modelling is correctly presented. Figures clearly detail the geometrical configurations that are considered, and the parameters investigated. The bibliography is extensive, reporting on a synthetic manner all the relevant results already published, for already established structures as well as for advanced 2D material structures.

This paper is a very good benchmark for the topic.

Response: Response: Thank you very much for your valueable feedback.

Reviewer 2 Report

The scope and organization of the manuscript is fine. I believe that the content of the manuscript is OK for publication. However there are some minor issues:

  1. the language of the manuscript can be improved better readability. for example the following lines require paraphrasing:
    • 57-58
    • 66-69
    • 80-82
    • 89-91
    • 200-201
    • 285-286
    • 386-387
  2. once the abbreviations are defined, stick to the abbreviations. For example in page to MS defined as metasurfaces. after that do not use metasurfaces any more. 
  3. Organization of section 6 should be consistent with figure 3. 
  4. in line 63, please provide some examples and references for "certain applications" 
  5. the sentence that starts at line 140 is not complete. 
  6. in line 147 "later"  should be replaced with "latter"
  7. in line 145-146 please elaborate and provide references for "The former approach has a disadvantage as in some bi-anisotropic metasurface, surface susceptibility has no physical meaning." 
  8. please avoid expressions like "as we all know" (line 209)
  9. in line 246-247 "In optics, they are implied in optoelectronics and displays" does not make sense.
  10. starting from page 10, figure labels are wrong.
  11. the sentence at line 386 requires modification. 
  12. line 467 and line 472 repeats same sentence for table 2. 
  13. caption of table 2 is wrong.

Author Response

Dear Respected Reviewer,

Nanomaterials,

We thank this valuable reviewer for the critical feedback. We feel lucky that our manuscript went to this reviewer as the valuable comments not only helped us to improve the manuscript but suggested some neat ideas for future studies. Please do forward our heartfelt thanks to these experts.  Based on the received comments, careful modifications have been made to the manuscript. All the changes have been made after track changes and are highlighted in yellow color in the manuscript. We hope our revised manuscript will meet the standard of Nanomaterials. Below, you will find point-by-point responses to the second reviewer’s comments and suggestions.

Please find the attached point by point response to the review Question for your kind consideration.

Reviewer 3 Report

This submitted munscript is a review of recent research on the Transmissive Polarizer Measurface. The first half of the paper provides an overview of its history and concept, and the second half presents specific studies. This is a useful review paper for researchers interested in this field.

However, there are many problems with the writing style of the paper. If the following points are corrected, the paper would be acceptable for publication.

1.There is an unnecessary underscore on line 48.

2.Could "i-e" in lines 67 and 68, 302, and 428 be a mistake for "i. e."?

3.Is "E.g." in line 103 a mistake for "e.g."?

4.The sentence in line 161, "Using [39] we can retrieve r1, r2, and 't', subsequently impedances Z1, Z2, and admittance Y can be Calculated." is too little explanation.

An explanation of what Z1, Z2, and Y mean in this system and what specific equations are derived should be included.

5.Due to lack of explanation, the equation in line 197 is not understandable. It should be explained what J and Y mean in this equation. Also, since the equation number is missing from this equation, the equation number should be added and all subsequent equation numbers in the text should be corrected as well.

6.Line 307 says "Whereas 'a' can be calculated from eq. (6) as:" but it is unclear why ‘a’ is calculated in this way from eq. (6). Also, the equation number is missing from line 308.

7. Line 348, the caption for Fig. 3 is too brief.

8.The figure numbers are messed up; isn't the figure on page 10 fig. 4? Subsequent figures are also mis-numbered. Also, figures (c), (d), (e), and (f) on page 10 are not cited in the main text and should be. the source of figure (f) on page 10 is not written.

9.With regard to the statement in lines 433 to 436, the supporting citation should be provided.

10. The figure on page 13 is not cited in the text and should be cited

。

Author Response

Dear Respected Reviewer,

Nanomaterials,

We thank this valuable reviewer for the critical feedback. We feel lucky that our manuscript went to this reviewer as the valuable comments not only helped us to improve the manuscript but suggested some neat ideas for future studies. Please do forward our heartfelt thanks to these experts.  Based on the received comments, careful modifications have been made to the manuscript. All the changes have been made after track changes and are highlighted in yellow color in the manuscript. We hope our revised manuscript will meet the standard of Nanomaterials. Below, you will find point-by-point responses to the third reviewer’s comments and suggestions.

Please find the attached point by point response to the review questions for your kind consideration.

Reviewer 4 Report

The submitted review “Transmissive polarizer metasurfaces: from microwave to optical regimes” by A.K. Fahad, C. Ruan and W. He is devoted to the state-of-the-art of design of metasurfaces with specific electromagnetic response, namely linear-to-circlular polarization converters. The basic known methods to make this structures frequency tunable are described. The review could be helpful for researchers and students. Nevertheless I think the text must be improved before the publication.

There are several misprints in the text, for example in line 48 is excessive “_” symbol, in line 102 “faraday’s” must be changed to “Faraday’s”, in line 141 it seems that two sentences must be united and separated by “,” not by “.”.

The numbering of figures and formulas must be checked and corrected (fig 2 in line 381 – must be fig 4, fig 3 line 413 must be fig 5 etc.) In line 197 formula is without number, in line 287 – formula has number 2, the same is the formula in line 151). Also I do not like the title for Figure 3 “Scheme for the article”, line 348. More informative title must be provided.

In my opinion section “Theory of Polarization Conversion” could be improved. Only the definitions are presented, like in formulas 2-4. The ways to calculate transmission matrix elements must be given (maybe on example of simple systems, like a surface of “not-coupled”  split ring resonators), or at least a text with explanations how it is usually done with references must be provided.

Also it sounds that the approaches in terahertz and optical and microwave and millimeter regions are different (lines 313-323). It must be explained what is the reason.  It is not clear whether the  φd in eq. 6 in line 306 is the same as á´“d in line 309 or maybe something else. This must be explained.

After this is fixed I think this review can be accepted for publication in Nanomaterials. 

Author Response

Dear Respected Reviewer,

Nanomaterials,

We thank this valuable reviewer for the critical feedback. We feel lucky that our manuscript went to this reviewer as the valuable comments not only helped us to improve the manuscript but suggested some neat ideas for future studies. Please do forward our heartfelt thanks to these experts.  Based on the received comments, careful modifications have been made to the manuscript. All the changes have been made after track changes and are highlighted in yellow color in the manuscript. We hope our revised manuscript will meet the standard of Nanomaterials. Below, you will find point-by-point responses to the fourth reviewer’s comments and suggestions.

Please find the attached point by point response to the review questions for your kind consideration.
